# A nonlinear polycrystalline ice elastoplastic contact model and its application

**Xiaolin Li[1,2], Ge Zhang[3,4], Xinyi Chen** **[5]***

1 Anhui Construction Engineering Quality Supervision and Inspection Station Co., Ltd, Hefei, China, 2 Anhui and Huaihe River Institute of Hydraulic Research, Hefei, China, 3 Key Laboratory of the Ministry of Education for Geological Hazards in the Three Gorges Reservoir Area (Three Gorges University), Yichang, Hubei, China, 4 School of Civil Engineering and Architecture, Three Gorges University, Yichang, Hubei, China, 5 School of Civil Engineering, Anhui Vocational and Technical College, Hefei, China

* xiaoxin3427@163.com

## Abstract

With the gradual warming of the global climate, the possibility and risk of large-scale sliding and collapse disasters of glaciers, large ice sheets, and thicker ice sheets have increased. Using the discrete element numerical method to analyze glacier stability is of great importance for polar disaster prediction. A contact model, which can accurately reflect the mechanical properties of polycrystalline ice, is key to conducting a discrete element numerical simulation of glacier stability. Based on the results of conventional triaxial compression tests on polycrystalline ice, we proposed an elastoplastic contact model. A custom contact model subroutine (dynamic link library [DLL]) for the two-dimensional particle flow code (PFC$^{2D}$) was generated using C++. A self-defined contact model subroutine (DLL) was used to simulate the biaxial shear of the flexible film at different temperatures. The numerical simulation results were in good agreement with the experimental results. The proposed contact model accurately reflected the deformation characteristics of polycrystalline ice. Finally, a discrete element numerical simulation of glacier ice collapse was conducted. An elastoplastic contact model of polycrystalline ice was established to provide a numerical basis for glacier stability analysis and multi-field coupling research.

## 1 Introduction

Glaciers and permanent snow are important components of the cryosphere, covering approximately 11% of the Earth's landmass. The terrestrial cryosphere is the world's largest reservoir of freshwater, accounting for approximately 75% of global freshwater resources [1]. China is the most developed country in the world in terms of glaciers at middle and low latitudes, with 48,571 existing glaciers and a total area of approximately 5.18e$^4$ km$^2$, accounting for approximately 0.54% of the country's national territory. China has glacier reserves of approximately 4.3–4.7e$^3$ km$^3$, making it second only to Canada, Russia, and the United States [2]. Currently, owing to

**Data availability statement:** All relevant data are within the manuscript and its Supporting Information files.

**Funding:** This study is funded by Yichang Natural Science Foundation(No. A23-2-007).

**Competing interests:** The authors have declared that no competing interests exist.

increasing warming of the climate, glaciers worldwide are in a state of severe retreat and material loss. The stability of structures and glaciers will be severely affected under the influence of earthquakes [3–5]. The deterioration of the mechanical properties of polycrystalline ice owing to temperature rise has led to an increasing number of geological disaster events, such as glacier destabilization and ice avalanches. The study of the mechanical properties of polycrystals is fundamental for understanding glacier movement and predicting geological hazards.

Since the 1930s, scholars worldwide have conducted extensive research on the mechanical properties of sea, glacial, and lake ice [6–14]. Through uniaxial and conventional triaxial compression tests, it was found that the compressive strength of ice is closely related to temperature, confining pressure, loading rate, and porosity [15–17]. Previous studies have shown that the compressive strength of ice increases with decreasing temperature, increasing confining pressure, and increasing loading rate. However, the tensile strength of ice is relatively less affected by temperature and strain rate. The average tensile strength of ice is approximately 1.43 MPa at a range of −10°C to −20°C [18–19]. Based on experimental data of the mechanical properties of ice, scholars have established strength criteria and constitutive ice models [20–23]. Finite element numerical simulations of glaciers are important. However, the finite element method cannot simulate the state of motion after ice avalanches and slides.

The discrete element numerical method abstracts real particles in the physical domain into particle units. Particle elements are used to construct specimen geometries. A contact model is used to perform interaction and iterative analyses to study the micromechanical properties of the structural specimens. Discrete element numerical methods have been widely used to study the macro–micro mechanical properties of geotechnical materials. Various contact models have been developed for different types of geotechnical materials [24–28]. Based on existing anti-rotation and anti-torsion models [24], Jiang et al. [29] established a contact model of loess that can comprehensively consider the coupled effects of water content, porosity, and absorption by introducing an inter-particle attraction to consider the effects of van der Waals and capillary forces. Shen et al. [30] used the discrete element method to simulate the mechanical response of cement between two spherical particles under tensile, compressive, shear, bending, torsional, and composite actions. A composite strength envelope and three-dimensional cement contact model were established. He et al. [31] introduced the softening relationship of a cohesive fracture model into a flat-joint contact model. The softening effect can then be considered when cohesive damage occurs between the particles.

Discrete element contact models for geotechnical materials have yielded numerous research results. However, the proposed contact models assume that the force–displacement relationship of the contact model remains linearly elastic until the contact is damaged. For materials such as polycrystalline ice and permafrost, the pressure–melting effect is very evident, which makes the force–displacement relationship of the contact nonlinearly elastic. Therefore, an elastoplastic discrete element contact model, considering the pressure–melting effect is proposed in this

study. The developed contact model is embedded into the two-dimensional particle flow code (PFC²ᴰ) software. The established contact model was used to perform numerical simulations of the biaxial compression of flexible polycrystalline ice membranes under different temperature conditions. The simulation results were then verified using experimental results.

## 2 Elastoplastic contact model

### 2.1 Introduction to contact models

The discrete element method considers a geometric material as consisting of a large number of discrete and independently moving rigid particle units, and the interaction between the particle units is controlled by the contact model. Based on Newton's second law, the velocity and position of the particles are constantly calculated and updated to obtain the kinematic morphology and mechanical behavior of the entire particle aggregate. Particle motion consists of two parts, translational and rotational. The control equation [32] is as follows:

$$m_i \frac{dV_i}{dt} = \frac{F_i}{m_i} + g \tag{1}$$

$$\omega_i = \frac{M_i}{I_i} \tag{2}$$

where $m_i$ is the mass of particle $i$, $V_i$ is the velocity of particle $i$, $g$ is the acceleration due to gravity, $t$ is the time, $\omega_i$ is the angular velocity of particle $i$, and $I_i$ is the rotational inertia of particle $i$, denoted as follows:

$$I_i = \frac{2}{5} m_i R_i^2 \tag{3}$$

where $R_i$ is the radius of particle, $F_i$ and $M_i$ are the combined force and moment on particle $i$, respectively, and are calculated as follows:

$$F_i = \sum_j \left( F_{ij}^n + F_{ij}^s \right) + F_i^{app} \tag{4}$$

$$M_i = \sum_j \left( F_{ij}^s r_{ij} \right) + M_i^{app} \tag{5}$$

where $F_{ij}^n$ and $F_{ij}^s$ are the normal and tangential forces between particles $i$ and $j$, respectively; $r_{ij}$ is the lit between the center of mass of the contact plane and the center of mass of particle $i$ and $F_i^{app}$ and $M_i^{app}$ are the forces and moments exerted on particle $i$ by the external load, respectively. Thus, it can be observed that the motion state of particle $i$ is closely related to the combined force and moment to which it is subjected. The force–displacement relationship of the contact model controls the updating of $F_{ij}^n$ and $F_{ij}^s$.

### 2.2 Discrete element elastoplastic contact model

When a discrete particle system is used to characterize a continuum medium, each discrete particle can be considered a unit. Two adjacent particles transmit force and moment through contact. At present, most discrete element contact models assume that the normal force–displacement and tangential force–displacement relationships between two particles are linearly elastic. For polycrystalline ice, the pressure–melting effect occurs under the action of external loads,

resulting in the normal force–displacement and tangential force–displacement relationships being nonlinear elastic. Displacements that cannot be recovered under the action of external loads are generated, namely, plastic displacements. In this study, a nonlinear elastoplastic discrete element contact model was established for discrete element simulation of polycrystalline ice.

**2.2.1 Normal force–displacement for elastoplastic contact models.** The normal force–displacement relationship of the contact is expressed in its full form as follows:

$$F_n = \frac{u_n}{n_a + n_b u_n}$$

(6)

The incremental form can be expressed as follows:

$$\Delta F_n = \frac{n_a}{(n_a + n_b u_n)^2} \Delta u_n$$

(7)

where $n_a$ and $n_b$ are the model parameters, $u_n$ is the normal displacement, and $F_n$ is the normal force.

The normal force–displacement relationship conforms to a hyperbola, as shown in Eq. (7), and the force tends to converge to a certain value with increasing displacement.

To reduce the parameters, it is considered that when the normal force is unloaded, the unloading stiffness is the initial stiffness $\frac{1}{n_a}$, and the unloading force–displacement curve is as follows:

$$F_n^{i+1} = F_n^i - \frac{1}{n_a} \Delta u_n$$

(8)

Then, the plastic displacement in the normal direction can be expressed as follows:

$$u_n^p = u_n - \frac{F_n}{(1/n_a)}$$

(9)

where $F_n^{i+1}$ is the normal force of the current step and $F_n^i$ is the normal force of the previous step.

Previous studies have demonstrated that polycrystalline ice has a certain and non-negligible tensile strength owing to ice crystal cementation. In this study, it was assumed that a certain tensile strength $t_n$ existed between the particle contacts. If the normal displacement is less than the plastic strain, the normal contact force is still calculated according to Eqs. (6) or (7). When the normal contact force is greater than the tensile strength $t_n$, the normal and shear contact force are equal to zero. A schematic diagram of the normal displacement–force relationship for the proposed elastoplastic contact model is shown in Fig 1.

The sensitivity analysis for the contact model parameters $n_a$ and $n_b$ (Fig 2) shows that the peak normal force increases as $n_a$ and $n_b$ decrease. The initial stiffness is proportional to $n_a$. The smaller $n_b$ is, the greater the displacement required for the curve to approach the asymptote.

**2.2.2 Tangential displacement–force relationships for elastoplastic contact modeling.** Similar to the normal displacement–force relationship, the tangential displacement–force relationship for particle contact satisfies the hyperbolic form. The incremental form of the tangential displacement–force relationship for particle contact is as follows:

$$\Delta F_s = \frac{s_a}{(s_a + s_b u_s)^2} \Delta u_s$$

(10)

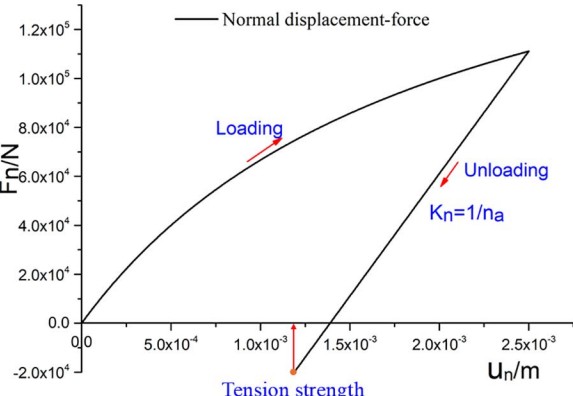

**Fig 1. Diagram of normal force–displacement relationship.**

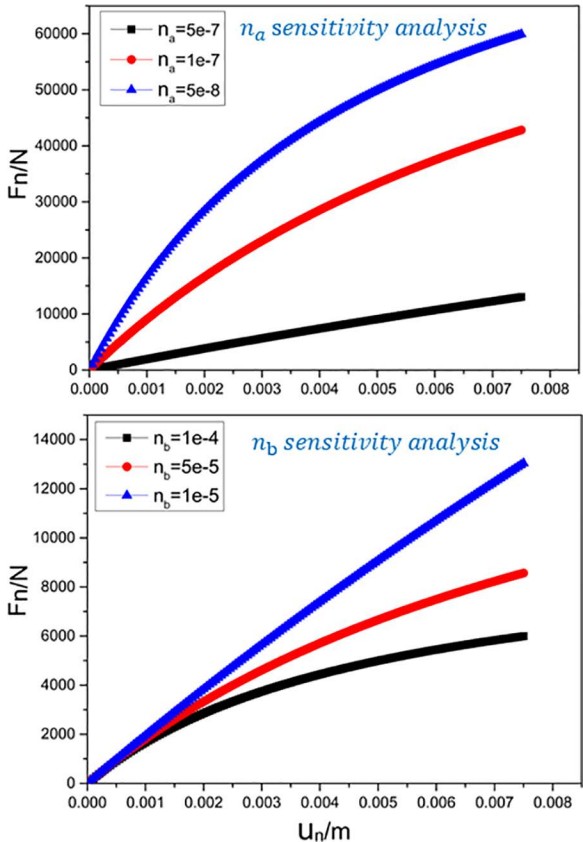

**Fig 2. Sensitivity analysis of normal contact relationship parameters.**

where $s_a$ and $s_b$ are the model parameters, $u_s$ is the tangential displacement, and $F_n$ is the tangential force.

At the same time, the tangential force needs to satisfy the Mohr–Coulomb strength criterion, and if $F_s > \mu F_N$, the tangential force is a sliding friction force (i.e., $F_s = \mu F_N$).

When the tangential force is unloaded, the unloading stiffness is always maintained at $\frac{1}{s_a}$ as follows:

$$F_s^{i+1} = F_s^i - \frac{1}{s_a}\Delta u_s$$

(11)

where $F_s^{i+1}$ is the tangential force of the current step and $F_s^i$ is the tangential force of the previous step.

Then, the plastic displacement in the tangential direction can be expressed as follows:

$$u_s^p = u_s - \frac{F_s}{(1/s_a)}$$

(12)

Reverse loading considers the tangential displacement–force relationship to be the same as the initial loading force–displacement curve and still satisfies the hyperbolic form. However, there is an irrecoverable plastic displacement in the tangential direction. Then, the tangential force–displacement relationship for reverse loading is expressed as follows:

$$\Delta F_s = \frac{s_a}{\left(s_a + s_b(u_s - u_s^p)\right)^2}\Delta u_s$$

(13)

The normal force–displacement relationship of the proposed elastoplastic contact model is shown schematically in Fig 3. The effects of the tangential model parameters $s_a$ and $s_b$ on the tangential force displacement are similar to those of the normal model parameters $n_a$ and $n_b$ on the normal force displacement, as described above.

## 3 Elastoplastic contact model development and validation

The PFC is developed based on the particle discrete element method [32], which simulates the deformation of an object using the force motion of a circular particle aggregate. Establishing contact model between particles is a key step in the particle discrete element method. The PFC framework provides a rich set of contact models and allows users to develop their contact models in C++. In this study, based on the contact model force–displacement relationship of polycrystalline ice, the source code of the contact model C++ program was completed and compiled to obtain a dynamic link library (DLL) file for the PFC^2D program.

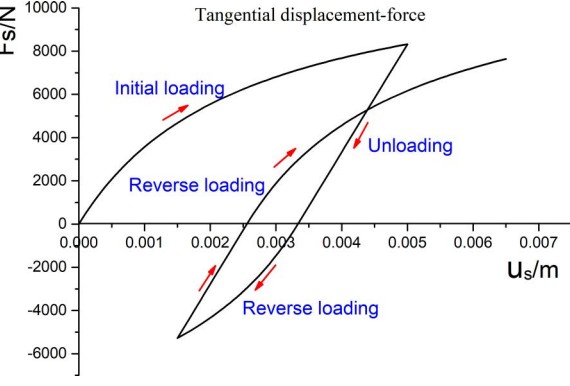

**Fig 3. Diagram of tangential force–displacement.**

### 3.1 Single-contact verification

A single contact was first used to show the normal force–displacement and tangential force–displacement laws. Whether the contact model is embedded correctly needs to be verified. As shown in Fig 4, two spherical particles with radii (R) of 0.05 m were generated. A single contact was produced between the particles with no overlap between them. The contact model parameters for the single contact were: normal parameter $n_a = 1.0e^{-8}$, normal parameter $n_b = 5.0e^{-6}$, tangential parameter $s_a = 1.0e^{-8}$, tangential parameter $s_b = 5.0e^{-6}$, friction coefficient $fric = 0.7$, and tensile strength $t_n = 1e^4$. The upper particle was fixed, and quasi-static loading was applied to the lower particle by applying a velocity in the y-positive direction. After some time, the lower particles were unloaded quasi-statically by applying a velocity in the y-negative direction. The normal force in contact and displacement of the lower particles were recorded. A comparison of the normal force–displacement curves obtained from the numerical simulation and the theoretical results are shown in Fig 4. The numerical simulation results agree with the predicted theoretical results.

To verify the correctness of the tangential force–displacement relationship, two spherical particles ($R = 0.05$ m) were generated, and a single contact between the particles was produced with an overlap between the particles of $D = 0.01$ m. The contact model parameters for a single contact were the same as those described above. The upper particles were then fixed, and the lower particles were loaded quasi-statically by applying a velocity along the x-positive direction. After some time, quasi-static unloading was performed by applying a velocity along the x-negative direction to the lower particles and reversing the loading. The tangential force of the contact and the displacement of the lower particle in the x-direction were also recorded. A comparison of the tangential force–displacement curves obtained from the numerical simulation and the theoretical results are shown in Fig 5. The numerical simulation results agreed with the predicted theoretical results.

In summary, it can be concluded that the elastoplastic contact model of ice was correctly written in the PFC$^{2D}$.

### 3.2 Numerical analysis of biaxial shear of polycrystalline ice

To further demonstrate the practicality of the established elastoplastic contact model, the proposed elastoplastic contact model was applied to carry out the conventional triaxial compression numerical simulation of polycrystalline ice under the conditions of a confining pressure of 1.0 MPa and temperatures of −2°C, −4°C, −7°C, and −15°C. The size of the model was 61.8 cm × 125 cm, and the total number of particles was 7733. The total number of contacts was 16814. The particle density was set to 900 kg/m³. The lower end of the specimen was fixed during the simulation, and the upper end was quasi-statically loaded using an axial strain rate constant of 1.67e⁻⁴(m/s). The simulation model is illustrated in Fig 6.

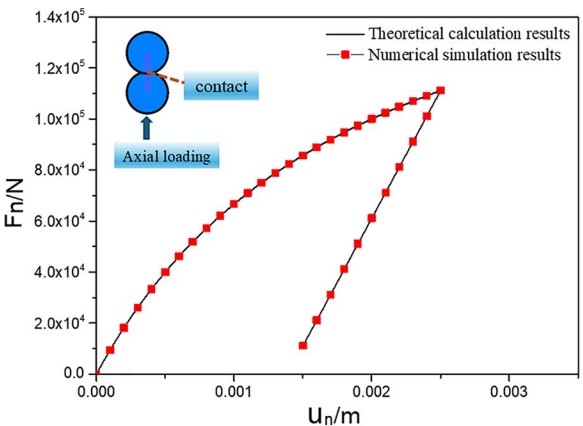

**Fig 4. Comparison between numerical simulation and theoretical results of normal contact force.**

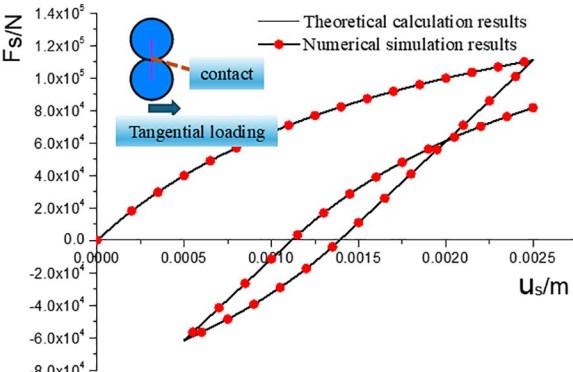

**Fig 5. Comparison between numerical simulation and theoretical results of tangential contact force.**

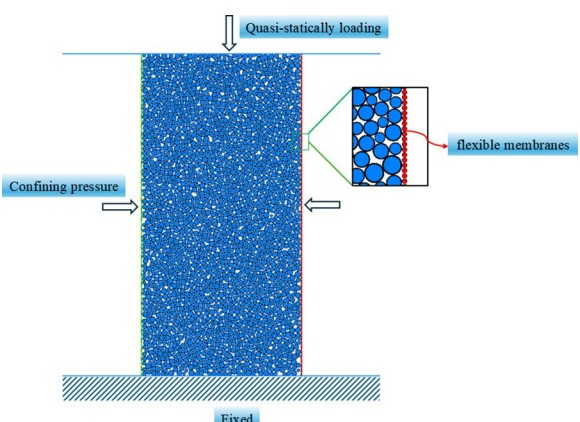

**Fig 6. Ice biaxial shear discrete element simulation model.**

Biaxial shear numerical simulations of flexible membranes at different temperatures (−2°C, −4°C, −7°C, and −15°C) were carried out using the proposed contact model. The obtained axial stress–strain curves, together with the results of conventional triaxial compression tests in the literature [15], are shown in Fig 7.

As shown in Fig 7, the simulation results of the proposed contact model agree with the experimental results, accurately reflecting the pressure–melting effect under axial loading. However, there are some errors between the numerical simulation and test results. There are several possible reasons for errors in discrete element numerical simulations. First, particles are often simplified to spherical shapes in discrete element numerical simulation, whereas actual particle shapes may be more complex. Second, it is typically assumed that the particles are rigid in DEM simulations, whereas the actual particles may be broken or worn, affecting their mechanical behaviors. Third, owing to computational resource constraints, the number of particles used in DEM simulations may be far fewer than the number of particles used in actual experiments, leading to differences in the statistical behaviors. Fig 8 shows the biaxial shear displacement cloud diagram of the flexible membrane at −7°C. It can be seen that a clear shear band appears near the top of the specimen. Thus, compared with existing models, the proposed elastoplastic contact model can better reflect the mechanical behavior of polycrystalline ice.

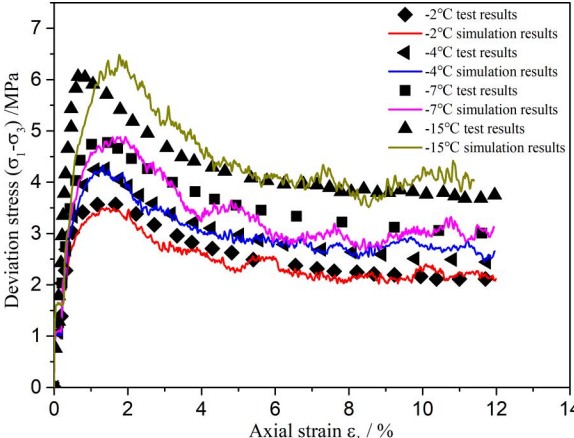

**Fig 7. Comparison between numerical simulation results and experimental results.**

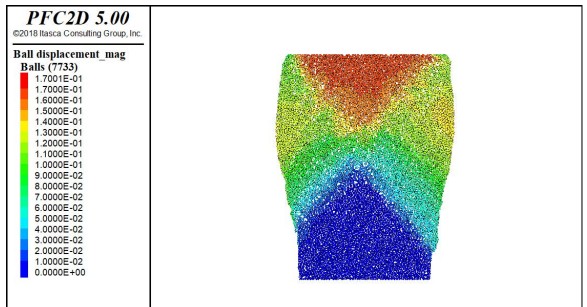

**Fig 8. Discrete element numerical simulation displacement diagram (−7°C for example).**

### 3.3 Sensitivity analysis of biaxial shear parameters for polycrystalline ice

To investigate the effect of the parameters of the customized contact model on the mechanical properties of polycrystalline ice, a sensitivity analysis of the biaxial shear parameters of polycrystalline ice was conducted for some of the parameters in the contact model. The results are shown in Figs 9–12. From the figures, it can be seen that as $n_a$, $n_b$, $s_a$, $s_b$ decrease, the stress–strain curves exhibit a transition from strain hardening to strain softening, with a gradual increase in the elastic modulus at the peak stress and at the initial stage.

## 4 Modeling of glacial destabilization processes

Between July 17 and September 21, 2016, two glaciers collapsed in the Ali region of western Tibet near Aru Co (Aru Lake). The massive glaciers first collapsed from the lower part of the Aru Glacier (34.03°N, 82.25°E) [33]. Gilbert et al. [34] obtained inverted bedrock topography using transient simulations, solving for meltwater infiltration and refreezing until the surface topography and enthalpy fields reached equilibrium with the climatic conditions. Yu [33] performed a numerical simulation of glacier instability based on inverted bedrock topography using the continuum–discontinuum numerical method. In the current study, we used the longitudinal section of the Aru glacier from the literature [33–34] to establish the model used for discrete element numerical simulation (Fig 13). The model contains a total of 19,018 discrete element particles, and the elastoplastic contact model proposed in this study was used in the contact model. Due to the lack of temperature

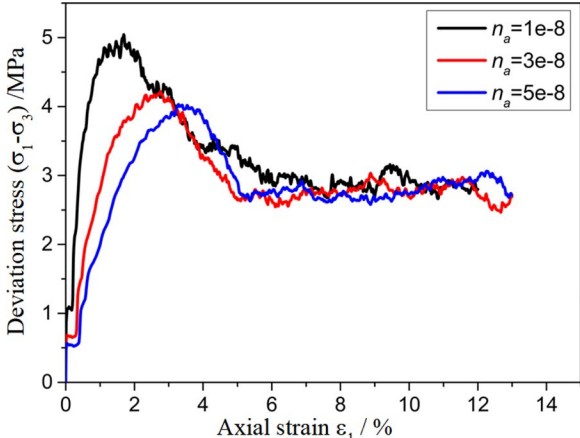

**Fig 9. Sensitivity analysis of model parameter $n_a$.**

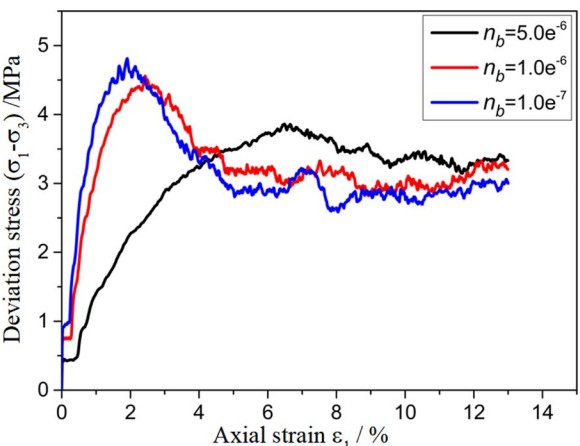

**Fig 10. Sensitivity analysis of model parameter $n_b$.**

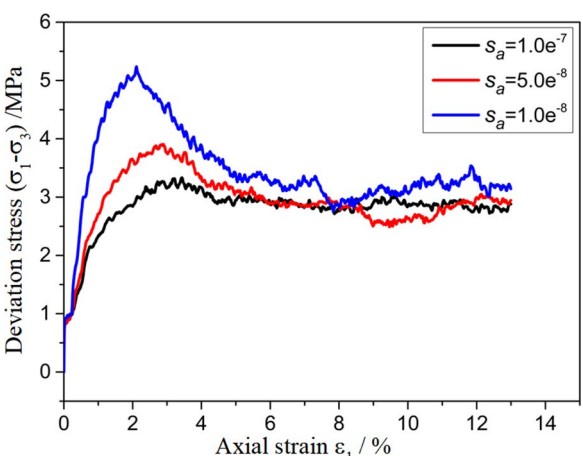

**Fig 11. Sensitivity analysis of model parameter $s_a$.**

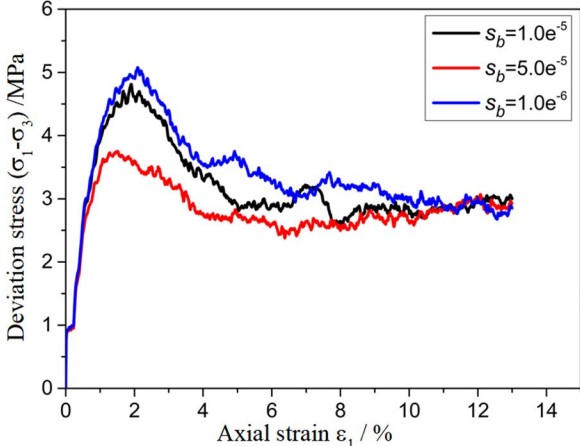

**Fig 12. Sensitivity analysis of model parameter $s_b$.**

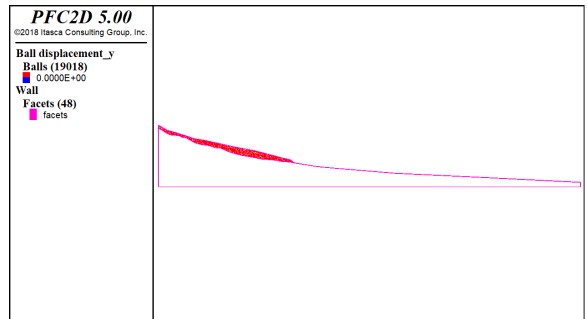

**Fig 13. Discrete element numerical simulation model.**

monitoring data for the Aru glacier, we assumed that the temperature is homogeneous inside the glacier. The contact model parameters are the same in all discrete elements, with normal parameter $n_a = 1.0e^{-8}$, normal parameter $n_b = 1.0e^{-7}$, tangential parameter $s_a = 1.0e^{-8}$, tangential parameter $s_b = 1.0e^{-7}$, friction coefficient $fric = 0.5$, and tensile strength $t_n = 1.0e^3$.

The contact between the polycrystalline ice and bedrock was modeled using a linear elastic contact model with the following parameters: normal stiffness $k_n = 1.0e^7$, tangential stiffness $k_s = 1.0e^7$, and friction coefficient $fric = 0.2$. The self-gravity stress is generated by the self-gravity of the glacier. The general tectonic stress is not considered. The time increment step of this calculation was set to $1.0e^{-5}$ s, and the total process of the numerically calculated ice avalanche was 200 s, with a total step length of 20000000 steps. To continuously react to the damage pattern of the ice body during the entire ice avalanche process, the results were simulated in accordance with certain time intervals. The cloud maps of the discrete element simulation of the glacial displacements for 0 s, 50 s, 100 s, and 200 s are shown in Fig 14. The discrete element numerical simulation of ice avalanches was performed using the developed elastoplastic contact model, which accurately reflected the entire ice avalanche process.

## 5 Conclusions

In this study, a discrete element elastoplastic contact model that can reflect the nonlinear force–displacement relationship of materials was proposed, and discrete element simulations of biaxial shear tests of polycrystalline ice flexible membranes were performed to verify the validity of the model. Our results provide a basis for subsequent discrete element

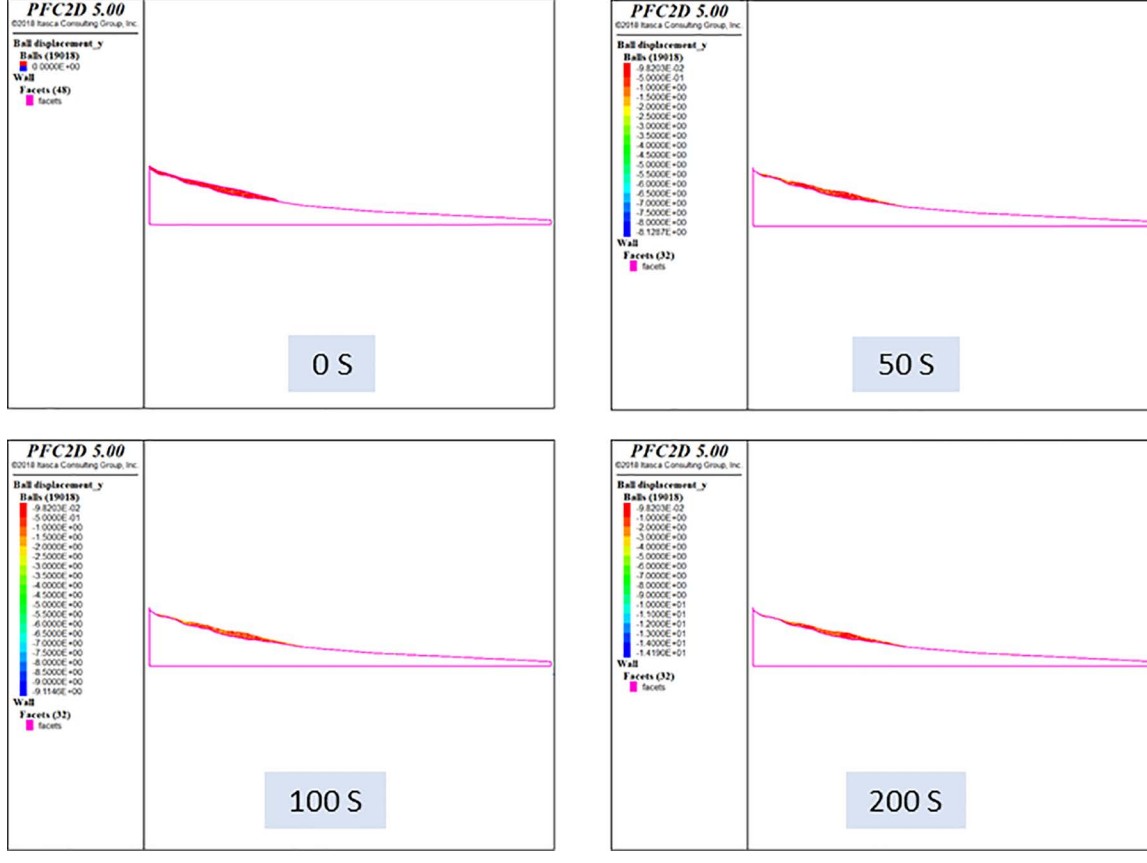

**Fig 14. Numerical simulation results of ice avalanche.**

numerical studies on the macroscopic and fine-scale properties of permafrost and polycrystalline ice. The main conclusions of this study are as follows:

(1) To consider the nonlinear behavior of some material contacts, such as that between frozen soil and polycrystalline ice, we proposed an elastoplastic model in which the contact force displacement is hyperbolic.

(2) The written contact model was validated using a single contact. The numerical simulation results were in perfect agreement with the theoretical calculations, and the elastoplastic contact model was considered to be correctly written.

(3) A numerical simulation of the biaxial shear of flexible membranes under different temperature conditions was performed using the PFC$^{2D}$ to develop a customized contact model. The simulation results agreed well with the experimental results, and the established elastoplastic contact model exhibited good practicability.

## Supporting information

**S1 File. Command stream.**
(ZIP)

**S2 File. Original data.**
(ZIP)

## Author contributions

**Funding acquisition:** Ge Zhang.

**Investigation:** Xiaolin Li.

**Project administration:** Xinyi Chen.

**Writing – original draft:** Xiaolin Li.

**Writing – review & editing:** Ge Zhang, Xinyi Chen.

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
