## [Decision Letter · Decision Letter 0]

Dear Dr. Chen,

Thank you for submitting your manuscript to PLOS ONE. After careful consideration, we feel that it has merit but does not fully meet PLOS ONE’s publication criteria as it currently stands. Therefore, we invite you to submit a revised version of the manuscript that addresses the points raised during the review process.

We look forward to receiving your revised manuscript.

Kind regards,

Dr. Guojin Qin

Academic Editor

PLOS ONE

“This study is funded by Yichang Natural Science Foundation(No. A23-2-007).”

6. We note that Figure 13 in your submission contain [map/satellite] images which may be copyrighted. All PLOS content is published under the Creative Commons Attribution License (CC BY 4.0), which means that the manuscript, images, and Supporting Information files will be freely available online, and any third party is permitted to access, download, copy, distribute, and use these materials in any way, even commercially, with proper attribution. For these reasons, we cannot publish previously copyrighted maps or satellite images created using proprietary data, such as Google software (Google Maps, Street View, and Earth). For more information, see our copyright guidelines: http://journals.plos.org/plosone/s/licenses-and-copyright.

a. You may seek permission from the original copyright holder of Figure 13 to publish the content specifically under the CC BY 4.0 license. 

Reviewers' comments:

Reviewer's Responses to Questions

**Comments to the Author**

1. Is the manuscript technically sound, and do the data support the conclusions?

Reviewer #1: Partly

Reviewer #2: Partly

2. Has the statistical analysis been performed appropriately and rigorously?

Reviewer #1: Yes

Reviewer #2: N/A

3. Have the authors made all data underlying the findings in their manuscript fully available?

Reviewer #1: Yes

Reviewer #2: No

4. Is the manuscript presented in an intelligible fashion and written in standard English?

Reviewer #1: Yes

Reviewer #2: Yes

Reviewer #1: It's interesting that this manuscript focuses on a nonlinear polycrystalline ice elastic-plastic contact model and its application,the research design is appropriate. What's more, the methods are adequately described and the results are clearly presented, so the conclusions are supported by the results. The analyses and the results reported in this manuscript are interesting and useful for future researches and preliminary design.

The following suggestions are needed to improve the scientificity and reliability of this manuscript.

1.There are problems with sentence structure, verb tense, and clause construction, the quality of English needs improving.It is noted that your manuscript needs careful editing by someone with expertise in technical English editing paying particular attention to English grammar, spelling, and sentence structure so that the goals and results of the study are clear to the reader.

2.Suggest reading these literature to expand research ideas, which may include (but are not limited to) the following, https://doi.org/10.1103/PhysRevB.54.3295.

https://doi.org/10.3311/PPci.15276. and https://doi.org/10.1016/j.soildyn.2018.03.021.

3.Figure 8 shows the biaxial shear displacement cloud diagram of the 231 flexible membrane at -7°C. Why choose a temperature of -7 °C?And why not any other numerical value?

After minor revision, it can be considered to be accepted.

Reviewer #2: The authors integrated the derived discrete element polycrystalline ice elastoplastic contact model into PFC 2D and conducted bidirectional shear numerical simulations with sensitivity analysis. Finally, the dynamic evolution based on PFC 2D was realized using a glacier collapse case in Tibet. This manuscript provides valuable insights for disaster prevention and response in engineering. However, insufficient structural logic, improper language use, vague expressions, and the low credibility of validation limit its overall quality improvement. Therefore, a major revision is needed. Specific suggestions are as follows:

(1) In the first section, lines 45 and 46, “5.18×104, 4.7×103” seem to be typos.

(2) It is recommended to enhance English readability throughout the manuscript.

(3) Do the equations in the second section originate from the authors? If not, please specify the references cited.

(4) In the second section, what is the basis for Fig.1 and Fig.2?

(5) In the second section, “Fn” in line 154 does not appear in Equation (10).

(6) In the second section, line 169, “normal displacement…”—does “normal” apply correctly in this context?

(7) In the third section, the validation in Sections 3.1 and 3.2 is solely based on subjectively generated cases, which may lack representativeness and generality. It is recommended to provide reasons for selecting the target case.

(8) In the third section, Fig.7 presents an experimental comparison with literature data and model validation. However, noticeable discrepancies are observed. It is suggested to add a discussion part on the sources of these errors.

(9) In the fourth section, lines 264–265, “this paper consider…is homogeneous”—what is the basis for this assumption?

(10) In the fourth section, Fig.15 is highly unclear, making it difficult for the reviewer to observe the glacier collapse process properly.

**Do you want your identity to be public for this peer review?** For information about this choice, including consent withdrawal, please see our Privacy Policy

Reviewer #1: No

Reviewer #2: No

---

## [Author Response · Author response to Decision Letter 1]

29 Mar 2025

Detailed Responses to the reviewers' comments

Ref: PONE-D-25-06117

Title: A nonlinear polycrystalline ice elastoplastic contact model and its application

The authors thank the editor and reviewers very much for their valuable comments. The manuscript has been extensively revised according to the reviewers’ comments, in which the places where the content has been revised are highlighted. The responses to the reviewers’ comments are also addressed in the manuscript, and the following is our response to the reviewers’ comments, item by item.

Comments from the Editor:

Response: Thank you very much for your suggestion. The manuscript is carefully reviewed to ensure that they meet the journal requirements.

2. Please note that PLOS ONE has specific guidelines on code sharing for submissions in which author-generated code underpins the findings in the manuscript. In these cases, we expect all author-generated code to be made available without restrictions upon publication of the work.

Response: Thank you very much for your suggestion. We have made our code open source.

“This study is funded by Yichang Natural Science Foundation(No. A23-2-007).”

Response: Thank you very much for your suggestion. The amended role of Funder statement has been added in the cover letter.

Response: Thank you very much for your suggestion. The data has been shared as an attachment.

Response: Thank you very much for your suggestion. I have completed the verification of my ORCID iD as required and ensured its successful association with my Editorial Manager account.

6. We note that Figure 13 in your submission contain [map/satellite] images which may be copyrighted. All PLOS content is published under the Creative Commons Attribution License (CC BY 4.0), which means that the manuscript, images, and Supporting Information files will be freely available online, and any third party is permitted to access, download, copy, distribute, and use these materials in any way, even commercially, with proper attribution. For these reasons, we cannot publish previously copyrighted maps or satellite images created using proprietary data, such as Google software (Google Maps, Street View, and Earth). For more information, see our copyright guidelines: http://journals.plos.org/plosone/s/licenses-and-copyright.

We require you to either (1) present written permission from the copyright holder to publish these figures specifically under the CC BY 4.0 license, or (2) remove the figures from your submission

Response: Thank you very much for your suggestion. We have removed the figure from the submission.

Comments from the reviewers:

Reviewer #1

1. There are problems with sentence structure, verb tense, and clause construction, the quality of English needs improving. It is noted that your manuscript needs careful editing by someone with expertise in technical English editing paying particular attention to English grammar, spelling, and sentence structure so that the goals and results of the study are clear to the reader.

Response: Thank you very much for your valuable suggestion. Your suggestions are of great significance to improve the quality of the article. We revised the language of the manuscript and invited specialized agency to polish the manuscript(Fig. 1).

Fig. 1. Editing Certificate

2. Suggest reading these literature to expand research ideas, which may include (but are not limited to) the following, https://doi.org/10.1103/PhysRevB.54.3295.

https://doi.org/10.3311/PPci.15276. and https://doi.org/10.1016/j.soildyn.2018.03.021.

Response: Thank you for your advice. We read these literature carefully. Those literature are very helpful to expanding our research ideas. We have added these papers to our manuscript.

3. Figure 8 shows the biaxial shear displacement cloud diagram of the flexible membrane at -7°C. Why choose a temperature of -7°C? And why not any other numerical value?

Response: Thank you very much for your advice. Because the biaxial shear displacement cloud diagram at -7°C is typical. It can well represent the displacement results at other temperature conditions. It can be seen from Fig. 8 that the shear band zone appears near the top of the specimen. The shear bond zone is an important phenomenon of polycrystalline ice behavior in the process of polycrystalline ice shear, which marks the transition from uniform deformation to local deformation. It is usually the precursor of soil failure. It can be proved that the proposed contact model can reflect the mechanical behavior of ice crystals well.

Reviewer #2

1. In the first section, lines 45 and 46, “5.18×104, 4.7×103” seem to be typos.

Response: Thank you very much for your advice. As you said, these are indeed the typos. We are deeply sorry for that. We have revised it in the text. The “5.18×104” should be “5.18×104”. The “4.7×103”should be “4.7×103”.

2. It is recommended to enhance English readability throughout the manuscript.

Response: Thank you very much for your advice. We have taken the time to review and revise our writing to ensure that it is clear and concise. To make the article easier to read, we have asked a professional polishing agency to help us polish the English writing(Fig. 1.).

3. Do the equations in the second section originate from the authors? If not, please specify the references cited.

Response: Thank you very much for your advice. The equations in the second section originate do not from the ours. They are originated from the help documentation of the PFC software. The relevant reference has been added to the manuscript.

4. In the second section, what is the basis for Fig.1 and Fig.2?

Response: Thank you very much for your advice. We apologize for the lack of clarity in the manuscript. In Fig. 1, the horizontal coordinate is the normal displacement, the ordinate is the normal force. During the loading process, the normal force is calculated according to Eq. (7), where the independent variable is the normal displacement increment. During the unloading process, the normal force is calculated according to Eq. (8). Two contact parameters(n_a, n_b) are included in Eq. (7). To study the influence of model parameters on the displacement- force relationship in the normal direction, the parameter sensitivity analysis is carried out.

5. In the second section, “Fn” in line 154 does not appear in Equation (10).

Response: Thank you very much for your advice. We are very sorry for our handwriting mistakes.

The “F_n” should be “F_s”. We have made a detailed examination of the meaning of the symbols in the manuscript.

6. In the second section, line 169, “normal displacement…”—does “normal” apply correctly in this context?

Response: Thank you very much for your question. We apologize for confusion caused by our lack of clarity. “the normal displacement-force relationship” means “the displacement- force relationship in the normal direction”. We have changed this inappropriate expression in the manuscript.

7. In the third section, the validation in Sections 3.1 and 3.2 is solely based on subjectively generated cases, which may lack representativeness and generality. It is recommended to provide reasons for selecting the target case.

Response: Thank you very much for your suggestion. Your suggestion is helpful to improve the quality of the manuscript. At the beginning, we put forward the theory of elastic-plastic contact model. Then, to carry out numerical simulation, we need to write the contact model into the pfc software by C++ programming. To verify that the programming is correct, a single contact verification is required. In Sections 3.1, the numerical simulation results are in agreement with the theoretical results. This shows that there is no problem with writing C++ code. In Sections 3.2, the proposed elastoplastic contact model is applied to carry out the conventional triaxial compression numerical simulation. This is to verify whether the proposed contact model can reflect the mechanical properties of polycrystalline ice. According to the experimental data[1], it can be seen that polycrystalline ice is a typical strain softening material. The triaxial numerical simulation results show that the proposed contact model can reflect the softening mechanical properties of polycrystalline ice well. It is proved that the proposed contact model can well reflect the mechanical properties of polycrystalline ice.

The detailed explanations are added to the manuscript.

Reference

[1] Xu, H. Y., Y. M. Lai, W. B. Yu, X. T. Xu, and X. T. Chang. 2011. “Experimental research on triaxial strength of polycrystalline Ice.” Journal of Glaciology and Geocryology. 33(5): 166-172.

8. In the third section, Fig.7 presents an experimental comparison with literature data and model validation. However, noticeable discrepancies are observed. It is suggested to add a discussion part on the sources of these errors.

Response: Thank you very much for your question. As you said, there are some errors between the numerical simulation and the test results. From the comparison results, it can be seen that the peak strength and residual strength are close. The peak strain corresponding to the peak strength has a certain error at -7°C. By referring to the relevant literature[2-3], it can be seen that there are some errors between the numerical simulation results and the test results. It proves that the errors in the manuscript are acceptable. There are several reasons for the error of discrete element numerical simulation. Firstly, particles are often simplified to spherical shapes in DEM simulation, while actual particle shapes may be more complex. Secondly, it is usually assumed that the particles are rigid in DEM simulation, while the actual particles may be broken or worn, affecting the mechanical behaviors. Thirdly, due to computational resource constraints, the number of particles used in DEM simulation may be far less than the number of particles in actual experiments, leading to differences in statistical behaviors.

The detailed explanations are added to the manuscript.

Fig. 1. Stress-strain curves and failure characteristics of frozen soil specimens without prefabricated hole for the tilting angle of 5◦ when subjected to impact loading[2]

Fig. 2. DEM simulation results under different conditions; (a) Grade A; (b) Grade B; and (c) Grade C[3].

[2] Lijun Zhang, Zhanfan Chunyu, Zhiwu Zhu, et al. DEM study on the impact mechanical properties and failure characteristics of frozen soil under coupled compression-shear loading, International Journal of Non-Linear Mechanics, 2025, 172, 105039.

[3] Dongyong Wang, Bo Shao, Jilin Qi, Wenyu Cui, Shengbin Jiang, Liyun Peng,Study on strain localization of frozen sand based on uniaxial compression test and discrete element simulation,

Cold Regions Science and Technology, 2024, 223: 104221.

9. In the fourth section, lines 264–265, “this paper consider…is homogeneous”—what is the basis for this assumption?

Response: Thank you very much for your question. As you say, polycrystalline ice in glaciers is bound to have some degree of heterogeneity. However, if consider the heterogeneity, it makes the calculation very complicated. The assumption will cause a certain deviation in the calculation, but it is acceptable. By referring to the relevant literature[4-5], it can be found that the heterogeneity of polycrystalline ice has been neglected in previous studies. In subsequent studies, we will consider the heterogeneity of polycrystalline ice.

[4] Yakubu Kasimu Galadima, Erkan Oterkus, Selda Oterkus, Peridynamic modelling of elastic and viscoelastic behaviour in polycrystalline ice: A study using NOSBPD and PDCHT, Ocean Engineering, 2024, 312(02), 119241.

[5] Dianzhe Li, Lu Liu, Yukui Tian, Shunying Ji,Numerical analysis on failure mode of 3D columnar polycrystalline ice based on discrete element method,Engineering Fracture Mechanics,Volume 315,2025,110837.

10 In the fourth section, Fig.15 is highly unclear, making it difficult for the reviewer to observe the glacier collapse process properly.

Response: Thank you very much for your suggestion. As you said, the clarity of this picture is really not high, we have modified it. we checked the other images as well.

---

## [Decision Letter · Decision Letter 1]

A nonlinear polycrystalline ice elastoplastic contact model and its application

PONE-D-25-06117R1

Dear Dr. Chen,

We’re pleased to inform you that your manuscript has been judged scientifically suitable for publication and will be formally accepted for publication once it meets all outstanding technical requirements.

Kind regards,

Guojin Qin

Academic Editor

PLOS ONE

Additional Editor Comments (optional):

Reviewers' comments:

Reviewer's Responses to Questions

**Comments to the Author**

Reviewer #2: All comments have been addressed

2. Is the manuscript technically sound, and do the data support the conclusions?

Reviewer #2: Yes

3. Has the statistical analysis been performed appropriately and rigorously?

Reviewer #2: N/A

4. Have the authors made all data underlying the findings in their manuscript fully available?

Reviewer #2: Yes

5. Is the manuscript presented in an intelligible fashion and written in standard English?

Reviewer #2: Yes

Reviewer #2: The authors have addressed the previous feedback and made improvements to the overall quality of the work. I therefore recommend accepting the manuscript for publication.

**Do you want your identity to be public for this peer review?** For information about this choice, including consent withdrawal, please see our Privacy Policy

Reviewer #2: No

---

## [Editor Report · Acceptance letter]

PONE-D-25-06117R1

PLOS ONE

Dear Dr. Chen,

I'm pleased to inform you that your manuscript has been deemed suitable for publication in PLOS ONE. Congratulations! Your manuscript is now being handed over to our production team.

Kind regards,

on behalf of

Dr. Guojin Qin

Academic Editor

PLOS ONE